https://doi.org/10.1038/s41467-019-09372-x | OPEN

# Humidity-tolerant rate-dependent capillary viscous adhesion of bee-collected pollen fluids

Donglee Shin[1], Won Tae Choi[1], Haisheng Lin[1], Zihao Qu[1], Victor Breedveld[1] & J. Carson Meredith[1]

We report a two-phase adhesive fluid recovered from pollen, which displays remarkable rate tunability and humidity stabilization at microscopic and macroscopic scales. These natural materials provide a previously-unknown model for bioinspired humidity-stable and dynamically-tunable adhesive materials. In particular, two immiscible liquid phases are identified in bioadhesive fluid extracted from dandelion pollen taken from honey bees: a sugary adhesive aqueous phase similar to bee nectar and an oily phase consistent with plant pollenkitt. Here we show that the aqueous phase exhibits a rate-dependent capillary adhesion attributed to hydrodynamic forces above a critical separation rate. However, the performance of this adhesive phase alone is very sensitive to humidity due to water loss or uptake. Interestingly, the oily phase contributes scarcely to the wet adhesion. Rather, it spreads over the aqueous phase and functions as a barrier to water vapor that tempers the effects of humidity changes and stabilizes the capillary adhesion.

[1] School of Chemical & Biomolecular Engineering, Georgia Institute of Technology, Atlanta, GA 30332, USA. Correspondence and requests for materials should be addressed to J.C.M. (email: carson.meredith@chbe.gatech.edu)

Bioadhesive systems often show useful qualities that have inspired many studies to elucidate their adhesive mechanisms[1–5] and mimic their functionality to improve the performance of synthetic adhesive systems[6–10]. For example, the adhesive pads on geckos suggest methods for creating strong and reversible dry adhesion[9], leading to mimicry of gecko adhesive functionalities[11,12]. The barbed proboscis of an endoparasitic worm swells to create strong mechanical interlocking after embedding in the soft tissue of its host[13], inspiring improved attachment reliability of microneedle patches on human skin[7]. Structural and experimental investigations of insect adhesive systems encourage the development of micropatterned adhesive tape for robotic applications[14]. While pollination transport is known to depend on adhesion and surface forces of the solids and liquids involved, very few studies have investigated the synergy between insect- and plant-derived liquid substances present on bee-collected pollen grains. Because of pollen size scales (from 10 nm features up to 100 μm diameter), insect rate of motion $(1–20 \, m \, s^{-1})$ and widely-varying temperature and humidity conditions under which these adhesives function, it is suspected that they could be a rich source of inspiration for new adhesive structures and functions. Their study could support an improved understanding of the role of these fluids in natural pollination and pollinator health.

Pollen is a significant nutrient source for bees[15] and their efficient collection and transport is essential for both the plant and animal's survival. Pollen grains are packed by the bees into pellets, carried on each hind leg in a structure called a corbicula or "pollen basket"[16]. Two full pollen pellets have a combined mass near 20 mg[17], which is more than 25% of the average body mass of the honey bee[18]. Reliable attachment of the heavy pollen pellets on the hind legs is essential for transport[19]. Honey bees utilize salivary secretions, or nectar, to adhere pollen grains together into a pellet and hold them on the corbicula hair[20,21]. However, many pollen grains also possess plant-derived coatings of pollenkitt, a complex mixture containing mainly saturated and unsaturated lipids, carotenoids, flavonoids, proteins, and carbohydrates[22]. The bees harvest pollen grains over a wide range of humidity conditions since the relative humidity in natural environments changes hourly and daily. Therefore, one may expect the liquid adhesive in the pellets to employ a mechanism to counteract changes in physical properties due to water loss or uptake associated with variations in humidity. Despite the importance of these fluids in pollinator and plant survival, the adhesive mechanisms of the liquid in the bee pollen pellets have not been studied extensively.

Maintaining the performance of synthetic adhesives after water uptake is a crucial challenge for adhesive applications[23], and synthetic adhesives can exhibit adhesion loss when a critical relative humidity is exceeded[24]. Evolutionary-adapted adhesive systems in nature show strategies for maintaining or enhancing adhesive properties under high relative humidity or even under water. For example, geckos[25] and ladybird beetles[26] show higher adhesion under a humid environment because absorbed moisture causes adhesive structures to swell and soften, leading to larger contact areas between their adhesive systems and rough surfaces. Some spiders utilize the moisture from the humid air to manipulate the viscosity of spider glue to maximize adhesion[27]. Furthermore, mussels[6], torrent frogs[28], and beetles[29] utilize structural or chemical properties of their adhesive systems to generate reliable underwater adhesion. An understanding of the adhesive mechanism and humidity dependence of the liquid secretion used in bee pollen pellets may support biomimicry strategies for humidity-tolerant synthetic adhesives.

Herein, we investigate the adhesive properties of the liquid secretion of dandelion pollen grains collected by honey bees (named as bee pollen adhesive in this paper). We observe that this adhesive forms a two-phase fluid, where the functions of each phase are unique. The aqueous phase is consistent with adhesive secretions generated in the honey stomach of bees[20,30], and the oily phase is consistent with pollenkitt, a plant-based oil coat on pollen grains[22]. We measure the wet adhesion of sunflower pollen (used as a control due to its regular surface features) with both bee pollen adhesive phases via colloidal probe microscopy. A surprisingly strong, rate-dependent adhesion occurs with the aqueous phase but not the oily phase. We also observe the ability of the oily phase coat (pollenkitt) to stabilize the physical properties of the aqueous adhesive relative to humidity changes. Without the pollenkitt, significant adhesion loss occurs for the aqueous phase at low and high relative humidity extremes. Adhesion loss is significantly curtailed (by a factor or more than 2) with the oily phase coat present.

## Results

**Two liquid phases of the bee pollen adhesive**. The pollen grains (Dandelion, *Taraxacum officinale*) collected by honey bees (*Apis mellifera*) (Fig. 1a–d) were purchased from Greer Laboratories (Lenoir, NC), stored in an unopened container at $-18 \, °C$ and used as received without further purification. The supplier uses traps at the entrance of hives to collect pollen from bees returning to the hive. Under these conditions, it is expected that pollens will be packed into a basket by the insect prior to being collected at the hive entrance in the pollen trap[20]. Bee pollen adhesive droplet samples were prepared as explained in the Methods section. Figure 1e shows an optical reflection-mode image of the droplet on a silicon wafer that indicates two phases are present. An uncolored, transparent core liquid region is surrounded by a distinct dark yellow liquid. To confirm that the two phases are chemically distinct, the Raman spectra of both regions (blue and red squares in Fig. 1e) were obtained using confocal Raman spectroscopy (Fig. 1f). The broadband peak between 3700 and $3000 \, cm^{-1}$ (O–H stretching) and intense peaks around 2941 and $2904 \, cm^{-1}$ (C–H stretching) were only observed in the core, and are typically detected in the Raman spectra of honey and sugars dissolved in water[31]. Honey bees utilize a mouth secretion composed predominantly of glucose and fructose dissolved in water to attach pollen grains to their pollen baskets[21]. There is no prior reported evidence of the presence of a yellow-pigmented separate phase in bee-derived bioadhesive. We hypothesize that the surrounding yellow region is pollenkitt, a plant-based lipid coat found on almost all entomophilous pollen[22] including dandelion[5]. The intense peaks (2669, 1520, 1150, and $1004 \, cm^{-1}$) of the Raman spectrum of the surrounding region are assigned to carotenoids, known components of pollenkitt[22,32]. Mixtures of flavonoids and carotenoids are yellow to orange in color[33], consistent with the yellow color observed in Fig. 1e, and these plant-based compounds are not known to be produced by bees. The Raman peaks assigned to the carotenoids showed smaller absorbance in the Raman spectrum of the aqueous phase (blue line in Fig. 1f), and they could be caused by a small amount of the oily phase dispersed in the aqueous phase as droplets, which is also evident in the microscopy images. While we expect that the fluids derive from pollen basket fluid, we cannot exclude the possibility that some isolated pollens are collected or that compounds from within the hive enter the pollen samples.

The liquid phase structure of the bee pollen adhesive droplet was investigated using confocal fluorescence microscopy in Fig. 1g. The top and side (in yellow box) view images were taken under 488 nm excitation, as one of the carotenoids (β-carotene) in pollenkitt is autofluorescent with excitation wavelength of 488 nm[33]. Both images show colloidal droplets containing carotenoids dispersed in the aqueous phase, but the main domain of the

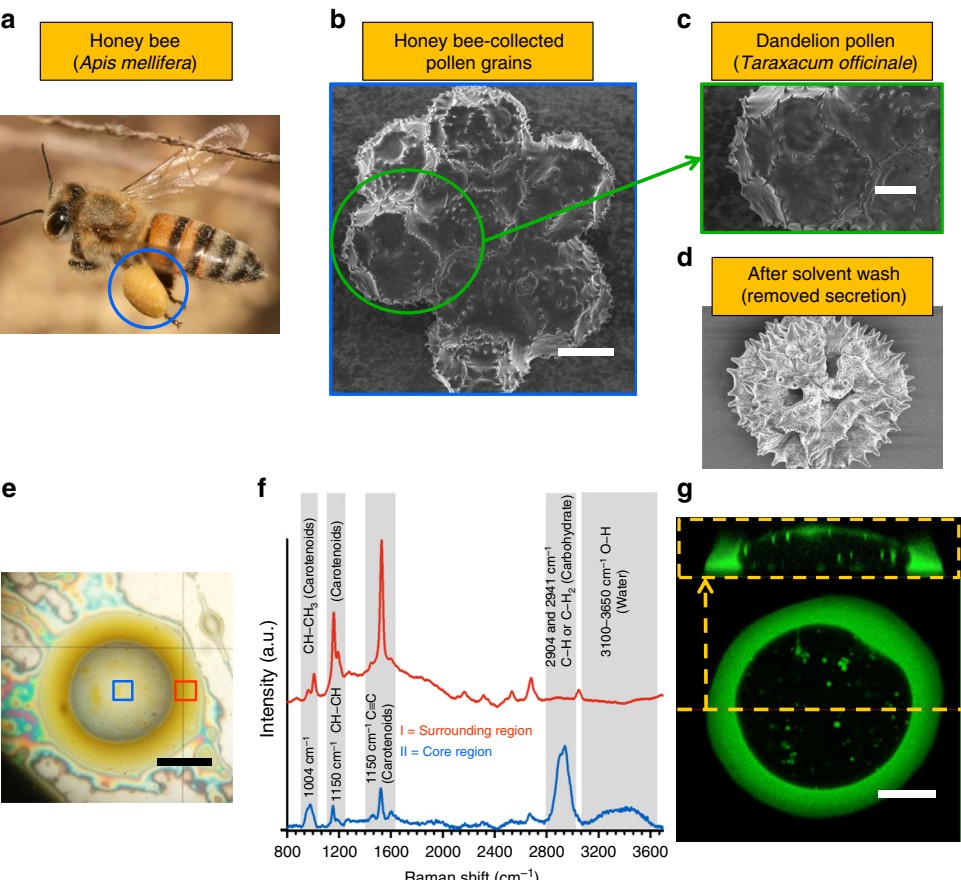

**Fig. 1** Two different liquid phases in bee-collected pollen adhesive. **a** Packed pollen grains (pollen pellet) on a pollen basket. Image is adapted with permission from Wikimedia Commons (photograph by Muhammad Mahdi Karim). SEM images of honey-bee-collected pollen grains (scale bar = 20 μm) (**b**) and a single dandelion pollen (scale bar = 10 μm) before (**c**) and after (**d**) solvent wash (water and toluene). **e** Optical microscope image of a bee pollen adhesive droplet on a silicon wafer (core region—blue box, surrounding region—red box). Scale bar = 60 μm. **f** Confocal Raman spectroscopy spectra of the surrounding (red) and the core (blue) regions of the droplet (488 nm laser excitation). **g** Confocal fluorescence microscopy images of a bee pollen adhesive droplet, top view with side view shown in a smaller in yellow box. Scale bar = 10 μm

aqueous phase is not fluorescent. On the other hand, the surrounding outer phase is predominantly fluorescent. The side view image shows that the aqueous phase droplet is coated with a thin oily film (~1 μm thickness). The droplet is determined by the spreading coefficient of the air–oil–water interface[34,35]. The observed structure, an aqueous phase drop cloaked by a thin oily film, indicates that the spreading coefficient of the oil phase at the air–aqueous phase interface is positive. While the evidence from the samples that were stored after collection (0 °C, 0.5% moisture) does suggest that the oily phase is pollenkitt and the aqueous phase has origin from bee secretions, a more formal confirmation of the ultimate source of the two fluids would require collecting fresh pollen directly from plants and bees and analysing it immediately without storage.

**Measurement of adhesive forces**. Adhesion of cleaned sunflower pollen (*Helianthus annuus*) on the core and surrounding liquid phases was measured using AFM colloidal probe microscopy at 20 °C and 40% RH, as described in Methods. It is desired to study the pull-out adhesion of a pollen particle in the liquid substances described above. However, the dandelion pollen has somewhat irregular features, making it difficult to reproducibly probe capillary adhesion of a single particle. Instead, we have utilized the sunflower pollen as a model, due to the fact that it comes from the same family as dandelion and possesses a regular ornamentation of well-defined conical spines. Both adhesion of

dandelion and sunflower have been studied on dry surfaces and shown to possess similar magnitudes, which derives from the similar chemistries of their exine[5]. The pollen attached on a tipless AFM cantilever was brought into contact with either the core or surrounding region of bee pollen adhesive droplets (Fig. 2a), and the pollen was retracted at different rates while the force was recorded as a function of distance. Typical force versus distance data are shown in Supplementary Fig. 1, and indicate features consistent with solid-liquid contact and capillary bridge formation during approach, as well as capillary thinning during retraction. The reported adhesion strength in subsequent figures is the value of the first minimum in the force-distance data, which corresponds to the maximum capillary adhesion achieved. For consistency, only force measurements in regions of liquid of similar thickness, in the range of 0.8–1.1 μm, are studied, significantly thinner than the sunflower pollen spine length of 4–5 μm. The thickness was estimated using the jump-in distance of the approach curves, as described in other studies[36,37].

The adhesion on the core region in Fig. 2b shows a remarkable rate dependence when the retraction rate exceeded 5 μm s⁻¹ but was independent of rate below 5 μm s⁻¹ (also visible in the raw force-distance data in Supplementary Fig. 1). The wet adhesion on the surrounding oil phase did not depend significantly on the retraction rate in Fig. 2b. Capillary forces and viscous dissipation are the two main contributors to adhesion of thin liquid bridges, and total adhesion is the sum of these forces[38]. The capillary force

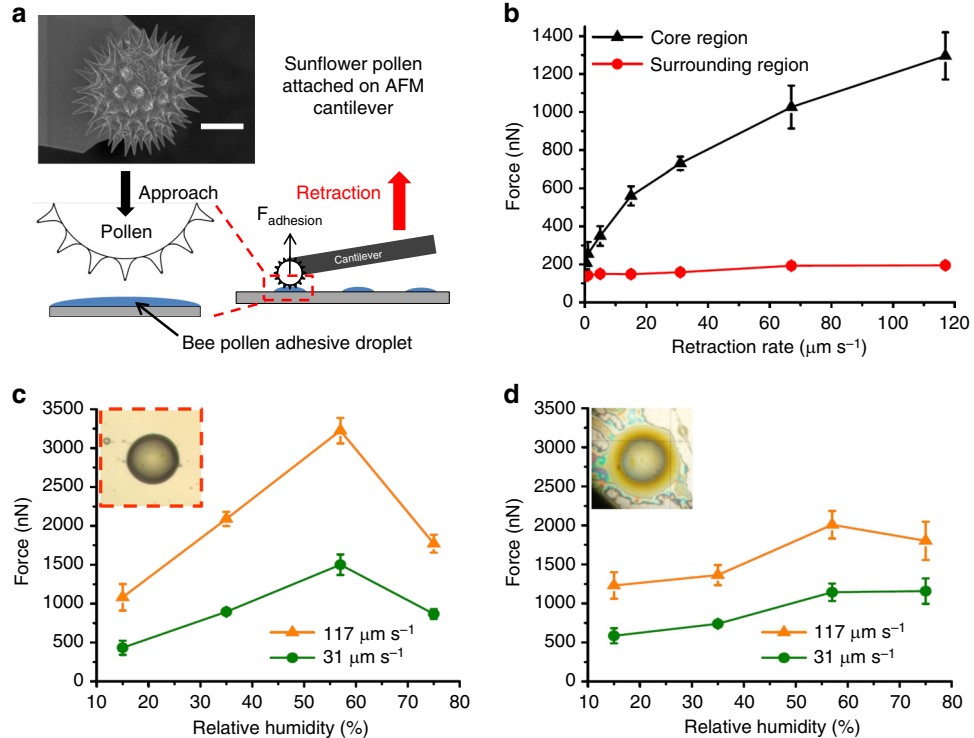

**Fig. 2** Rate- and humidity-dependent adhesion of bee pollen adhesive. **a** Schematic of adhesive force measurement of cleaned sunflower pollen with bee pollen adhesive fluid using AFM colloidal probe microscopy. Scale bar = 10 μm. **b** Adhesive force versus retraction rate of a sunflower pollen (*Helianthus annuus*) on the core (mainly aqueous phase) and the surrounding (oily phase) liquid regions of a bee pollen adhesive droplet. The error bars are s.e.m. **c**, **d** Adhesive force of a sunflower pollen on the aqueous phase droplets with oil phase removed (**c**) and the oil phase coat intact (**d**), for bee pollen adhesive stored at different relative humidity levels and at two retraction rates of 31 and 117 μm s⁻¹. The error bars are s.e.m.

is a static component of adhesion deriving from surface tension and Laplace pressure forces. The viscous dissipation arises due to hydrodynamic response of fluid in the capillary bridge and its magnitude is dependent on separation rate and viscosity[37]. The core region displayed a drastic increase in adhesion force with increasing retraction rate while little change in adhesion was observed in the surrounding region. The adhesion magnitude generated in the core (207 ± 33 nN) was 49% stronger than adhesion on surroundings (139 ± 4 nN) at a retraction rate of 0.5 μm s⁻¹. However, adhesion of the core increased to 660% higher than surroundings at a rate of 117 μm s⁻¹. The large magnitude of adhesion on the core aqueous phase relative to the surrounding oily fluid suggests that the aqueous phase of bee pollen adhesive fluid is the main contributor to adhesive properties, and this aqueous phase adhesion is mainly attributed to viscous dissipation during capillary thinning, demonstrated below.

Bee pollen adhesive droplets resting on a silicon wafer were washed with toluene, as described in Methods. After washing, the surrounding oily phase was clearly removed (red box in Fig. 2c), and no notable shape or size change of the core region (aqueous phase) was observed. The washed samples were utilized for investigating the wet adhesive properties of the aqueous phase. Before the force measurement, the thickness of the aqueous fluid was measured as described in the Methods section. Three aqueous phase droplets (total 12 droplets), each with a thickness in the range 1.5–1.8 μm, were chosen for the wet adhesion measurement as shown in Supplementary Fig. 2 and were kept at the same relative humidity (15, 35, 57, and 75% RH) for 24 h prior to measurement.

In Fig. 2c, the maximum adhesive force of the aqueous phase decreased by more than 40% when the relative humidity was increased from 57 to 75% RH. In addition, a decrease in adhesion

was observed when transitioning from 57% RH to lower relative humidity (15 and 35% RH). A similar variation of wet adhesion magnitude, with a maximum in adhesion at a certain RH, was also observed for spider viscid glues[27]. This dependence of adhesion magnitude on RH was explained by counteracting effects of viscosity on the viscous contribution to capillary adhesion and the spreading rate of the adhesive[27]. Because glucose and fructose are well-known water absorbing materials, we suggest that this competition is also responsible for the observed humidity-dependent adhesion behavior in the aqueous phase of bee pollen adhesive. As a result, the viscosity of the aqueous phase is expected to decrease[39]. The force magnitude from viscous dissipation during capillary thinning is proportional to the viscosity of the liquid[38], consistent with the reduction of liquid viscosity observed at the highest relative humidity in Fig. 2c. The wet adhesion at low relative humidity (15 and 35% RH) was also decreased relative to that at 57% RH, even though the viscosity of the samples increases as water desorbs[39]. The adhesion loss is attributed to the reduced spreading rate of liquid adhesive of higher viscosity. The normalized spreading radius of the aqueous phase for 10 s (reported in Supplementary Fig. 3) shows a drop, from 1.3 at 75% RH to 1.0 at 30% RH. All adhesion AFM measurements were performed with the same approach rate of pollen to the droplet (0.5 μm s⁻¹), so that the contact period for formation of a capillary bridge during the approach step was constant (~3 s). An increasingly smaller wetting area will result as viscosity increases at lower humidity, leading to a capillary bridge with smaller volume. As a result, the magnitude of the capillary viscous, and static, contributions will decrease.

We have shown that the adhesion loss of the aqueous phase can be attributed to water uptake or loss of the bee pollen adhesive at high or low humidity, respectively. Consequently, if

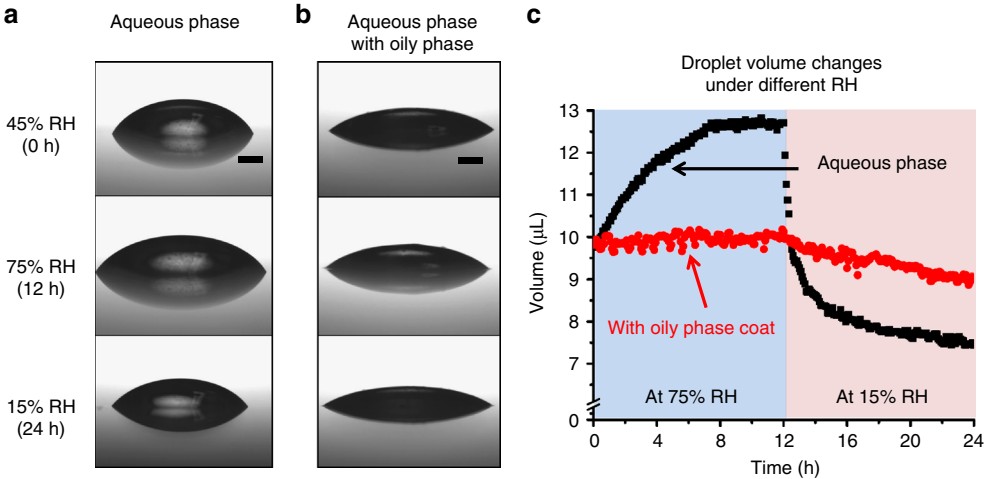

**Fig. 3** Water uptake and loss of bee pollen adhesive stored in high and low humidity environments. Images of the aqueous phase droplet (aqueous extract) without oily phase coat (toluene extract) (**a**) and the aqueous phase droplet with the oily phase coat (**b**) on a polystyrene surface, stored at 75% RH for the first 12 h and at 15% RH for the next 12 h. The droplet was initially saturated at 45% RH for 2–3 days. The black scale bars represent 1 mm. **c** The measured droplet volumes are shown as a function of time

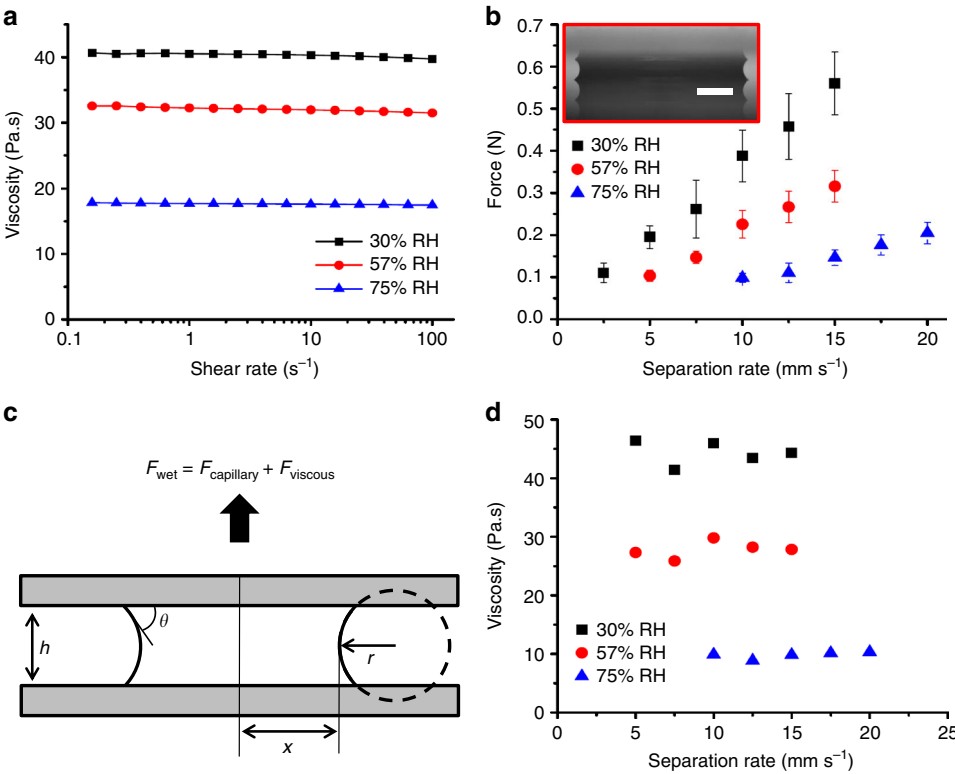

**Fig. 4** Influence of humidity-dependent viscosity changes on the macroscopic wet adhesive performance of the aqueous extract fluid. **a** Viscosity of the aqueous extract samples stored at different relative humidity levels (30, 57, and 75% RH for 3 days) as a function of shear rate at 20 °C. **b** Adhesion force due to the aqueous extract liquid bridge between two glass slides (as shown in red box, scale bar = 1 mm) as a function of separation rate and humidity. The error bars represent 95% confidence interval. **c** Schematic of the separation of two parallel flat surfaces joined by a liquid bridge. **d** Viscosity of the aqueous extract stored at different humidity levels as estimated by the wet adhesion model (Eq. (4)) by using the measured liquid-bridge forces (Fig. 4b)

the water loss or gain of the aqueous phase is attenuated by the pollenkitt oily phase, the aqueous-core adhesive's sensitivity to changes in humidity can be decreased. We measured the adhesion of sunflower pollen on the bee pollen adhesive droplets stored at different relative humidity levels. Before the force measurement, the thicknesses of the bee pollen adhesive droplets were 1.1–1.3 μm as shown in Supplementary Fig. 4. The results, shown

in Fig. 2d, indicate that the aqueous phase coated with oily phase shows a strong adhesion at 57% RH (Fig. 2d), like the adhesion of the toluene-washed aqueous phase in Fig. 2c, but the adhesion loss was attenuated at low (15% RH) and the high (75% RH) relative humidity, compared to the case of the uncoated aqueous phase in Fig. 2c. The adhesion loss of the samples with the oily phase was only half of the reduction of the aqueous-only phase at

**Table 1 Dimensions of the liquid bridge of the aqueous extract stored in different humidity levels**

|        | Azimuthal radius (x, mm), | Meridional radius (r, mm) | Height (h, mm) | Contact angle (θ, °) | Estimated static capillary force (mN) |
|--------|---------------------------|---------------------------|----------------|----------------------|---------------------------------------|
| 30% RH | 2.68 ± 0.08               | 0.34 ± 0.02               | 0.66 ± 0.01    | 26.2 ± 2.3           | 4.3                                   |
| 57% RH | 2.60 ± 0.07               | 0.36 ± 0.01               | 0.66 ± 0.01    | 36.7 ± 3.1           | 4.0                                   |
| 75% RH | 2.81 ± 0.07               | 0.34 ± 0.01               | 0.67 ± 0.03    | 36.3 ± 4.2           | 4.9                                   |

The errors represent 95% confidence interval

both low (15%) and high (75%) relative humidity from the adhesion at 57% RH. Thus, the oily phase appears to attenuate changes in adhesion.

**Water absorption and dehydration.** Above, we showed that the aqueous phase was diluted by absorbing moisture from humid air, or concentrated by evaporating water in dry air, resulting in changes to wet adhesion behavior. The aqueous and oily phase fluid in bee pollen adhesive were extracted and separated using solvent extraction with water and toluene as described in Methods (named as aqueous extract and toluene extract, respectively), and their physical properties are shown in Supplementary Table 1. After solvent extraction, the Raman spectra of the aqueous and toluene extracts were obtained using confocal Raman spectroscopy (Supplementary Fig. 5), and they were closely similar to the core and surrounding regions (Fig. 1f). The viability of pollen is quickly decreased with dehydration[22], and pollen grains from some plant species are thought to use the pollenkitt to reduce water loss. Thin films of the oily phase spread over the aqueous phase could serve to reduce water uptake under elevated humidity and reduce water loss under reduced humidity, helping to mitigate the effects of humidity on the bee pollen adhesive. The volume changes of an aqueous phase droplet (aqueous extract) with an oily phase (10 wt%) coat (toluene extract) was tested to confirm the role of the oily phase, and the results are shown in Fig. 3a, b. The aqueous extract was initially stored at 45 ± 5% RH for 2–3 days. As shown in Fig. 3c, the volume of the aqueous extract droplet was increased by 30% from the original volume when it was exposed to a high humidity environment (75% RH) for 12 h. The volume decreased by 24%, compared to the initial droplet volume when the humidity decreased to 15% RH for 12 h. This result indicates that the water content of the aqueous extract is strongly dependent on the relative humidity of surrounding air, and supports the prior hypothesis that adhesive property changes of the aqueous phase can be attributed to the water uptake or loss.

**Viscosity measurement.** The viscosity of the aqueous extract fluid stored at different relative humidity levels was measured in a rheometer to understand the influence of the water uptake or loss, as described in the Methods section. Figure 4a shows that the viscosity of the aqueous extract is strongly dependent on the relative humidity, as expected. The viscosity increased from 32 to 40 Pa·s when humidity decreased from 57 to 30% RH, but reduced to 18 Pa·s at higher humidity (75% RH). These changes in viscosity were consistent with a notable increase in sample volume as a function of increasing RH. The aqueous extract is Newtonian as shown in Fig. 4a, consistent with prior rheological studies of honey and sugar solutions[40].

In this section, we discuss how the humidity dependence of viscosity affects the wet adhesive mechanism of the bee pollen adhesive aqueous phase. To eliminate the influence of differences in spreading rate of fluid during approach and geometrical factors associated with microscopic pollen and droplets, we utilized a macroscopic adhesion test based on the separation of two prewetted parallel flat surfaces connected by the aqueous extract

fluid. A droplet of 15.8 ± 0.2 μl of the aqueous extract was placed on a glass slide, and a liquid bridge of the aqueous extract was formed in between the two flat glass surfaces as shown in the red box in Fig. 4b. Multiple capillary bridge samples were prepared in the same manner and stored in different relative humidity chambers (30, 57, and 75% RH) for 24 h before the adhesion measurement. The dimensions of the capillary bridges were measured using a goniometer immediately before the test. A custom load-displacement sensing apparatus[41] was used to measure the adhesion magnitude of the liquid bridges with separation rates in the range of 5–20 mm s$^{-1}$ (reported in Supplementary Fig. 6), and measured force-time curves are shown in Supplementary Fig. 7. As shown in Fig. 4b, we observed that the adhesive force magnitude from the capillary bridges was very sensitive to the relative humidity, and the adhesion shows a nearly linear relationship with the separation rate. Wet adhesion models for the separation of two parallel flat surfaces[38] were used to understand these relationships. Capillary and viscous forces are two main contributors to the adhesion of thin liquid bridges, and the total adhesion of liquid bridges can be estimated by the summation of these forces. The viscous and static contributions to the wet adhesion can be determined by considering the meniscus curvatures, dynamics and viscosity[38]. The capillary force is caused by a liquid bridge between the two separated surfaces, and the curvature of the liquid meniscus is characterized by two radii which are the azimuthal radius ($x$) and the meridional radius ($r$), indicated in Fig. 4c. The physical configuration of the wet adhesive model is shown schematically in Fig. 4c. The total capillary force can be defined as the summation of the surface tension and Laplace pressure contribution[42] as shown in Eq. (1),

$$F_{\text{capillary}} = 2\pi\gamma x \sin\theta + \Delta P\pi x^2, \qquad (1)$$

where $\gamma$ is the surface tension of the liquid, $\theta$ is contact angle between the liquid bridge and the flat surfaces, and $\Delta P$ is Laplace pressure, estimated by the Young-Laplace equation:

$$\Delta P = \gamma\left(\frac{1}{x} + \frac{1}{r}\right) \qquad (2)$$

The surface tension ($\gamma = 52.5 \pm 0.1$ N m$^{-1}$ at 57% RH) of the aqueous extract fluid was measured using drop shape analysis of a pendant drop. The azimuthal radius ($x$), the meridional radius ($r$), the height ($h$), and contact angles ($\theta$) of the liquid bridges were determined from optical images, taken immediately before the force measurement, as shown in Table 1.

Based on the measured configurations, the static capillary forces of the liquid bridges were estimated by Eq. (1) as shown in Table 1. The estimated static capillary forces were two orders of magnitude lower than the measured adhesion in Fig. 4b. Even if one uses the pure water surface tension (72.8 N m$^{-1}$), the estimated static capillary contribution is much smaller than the measured forces in Fig. 4b. These results indicate that the viscous force governs the wet adhesion magnitude of the aqueous extract, a common phenomenon when the liquid bridge is a highly viscous liquid[43].

The viscous force is an effect of the hydrodynamic response of the liquid bridge, resisting the separation of the two flat surfaces. The viscous force of a Newtonian fluid for a non-slip boundary condition can be predicted by the Reynolds lubrication equation. The average pressure difference, caused by the hydrodynamic response, between the liquid bridge and ambient air during the separation was estimated previously by Cai and Bhushan (2007):[38]

$$\Delta P_{avg} = \frac{3\eta}{2h^3} x^2 \frac{\partial h}{\partial t}, \qquad (3)$$

where $\eta$ is the viscosity of the liquid, and $\partial h/\partial t$ is the separation rate. Therefore, the viscous force of the liquid bridge having a meniscus area of $\pi x^2$, at a constant separation rate $\dot{h}$, can be approximated as

$$F_{viscous} = \frac{3\eta\pi}{2h^3} x^4 \dot{h} \qquad (4)$$

The aqueous extract shows a Newtonian fluid behavior as shown in Fig. 4a, so the viscous force and the separation rate ($\dot{h}$) should have a linear relationship with the slope of $3\eta\pi x^4/(2h^3)$ as shown in Eq. (4). This linear relationship was observed in the measured adhesion (Fig. 4b), from which the viscosity of the aqueous extract samples was estimated by using Eq. (4). This result, taken with the viscosity and water absorption data presented previously, indicates that the viscosity changes induced by variations in environmental humidity appear to be the main cause of the humidity-dependent adhesion of the aqueous phase. This also supports the prior explanation of humidity-dependent viscosity causing the observed decrease in capillary adhesion of pollen and bee pollen adhesive at high humidity. The viscosity estimated from the adhesion results (Fig. 4d) agrees reasonably with the viscosity measured by a rheometer in Fig. 4a. Quantitative differences in the rheologically-determined and adhesion-determined viscosity values were possibly related to the different volumes of the samples and water saturation in the different instruments. The volumes of the aqueous extract used to measure the forces (15.8 μl, saturated for 24 h) were much less than the volume used for the rheometer tests (1 ml, saturated for 3 days). In addition, the viscosity measured in a rheometer was quite sensitive to temperature changes in the range of 20–40 °C, as shown in Supplementary Fig. 8, and the discrepancies were probably attributed to the temperature-sensitive property of the samples, as well.

## Discussion

This study has revealed for the first time the presence of two immiscible liquid phases in a bioadhesive (called 'bee pollen adhesive' in this paper) extracted from the honey-bee-collected pollen grains. In previous studies, bee mouth secretion was recognized as a sugar-based aqueous phase bioadhesive to hold pollen grains on their pollen baskets[20,21]. We have demonstrated the presence and function of the oily phase (likely to be pollen-kitt) from pollen in the bee pollen adhesive. The oily phase spreads on the aqueous phase and significantly tempers the influence of humidity on the aqueous phase water content. The adhesion of the aqueous phase without the oily phase was reduced to under half (low RH) or near half (high RH) of the maximum adhesion observed at intermediate humidity (57% RH). However, the magnitude of the adhesion loss at low and high RH decreased by about half of the original reduction when the adhesive samples were coated by the oily phase. The oily phase served to prevent excessive water absorption under elevated humidity and excessive drying under reduced humidity.

Viscous hydrodynamic adhesion and stabilizing humidity dependence play a significant role in establishing the interesting properties observed for this two-phase naturally-derived material. The adhesion magnitude of bee pollen adhesive shows a rate-sensitive response, and this is a relatively unexplored feature in particle or capillary adhesion that can be utilized to generate a tailorable force magnitude. This fascinating tailorability could be utilized potentially to control motion in microscopic or nanoscale devices, in environmental remediation or particle abatement. In addition, these two adhesive functions are active at macroscopic scales as well, suggesting bioinspired rate- and humidity-tunable adhesion applications in construction, medicine, and other fields. The rate-dependence could be functionally useful for enabling the use of capillary forces for controlled displacement or transfer of microscopic parts, i.e., MEMS, assembly, or printing. Perhaps the rate dependence serves a role in collection and transfer of pollen by insects; however, further studies should be carried out comparing pollen freshly collected from plants (without bee secretion) and pollen freshly collected from bees, to investigate the biological implications. The oily phase coat on bee pollen adhesive shows a distinctive functionality to preserve water content. This action stabilized the viscosity of the aqueous phase during humidity changes, thereby stabilizing the hydrodynamic adhesion contributed by the aqueous phase. The results provide inspiration for the future development of novel humidity-stabilized adhesive materials based on the formation of a liquid water-barrier external oil phase layer.

## Methods

**Bee pollen and bioadhesive sample preparation.** The bee-collected pollen grains (Dandelion, *Taraxacum officinale*) were purchased from Greer Laboratories (Lenoir, NC), stored in an unopened container at −18 °C and used as received without further purification. The supplier certifies via microscopic analysis the amount of other plant parts present is 2.8% and the contamination by foreign pollen, mold or rust is 0.01%. For storage, the collected pollen grains were dried to 0.5% moisture content, and stored at <0 °C. To collect sufficient quantities of bee pollen adhesive fluid, 35 mg of dandelion pollen was deposited on a piranha-etched silicon wafer and held at 20 °C (57% RH) for 24 h to allow time for the fluid to drain. The pollen grains were subsequently removed by blowing with nitrogen gas, leaving bee pollen adhesive droplets on the silicon wafer. These droplets were used for characterizations described below. To isolate the aqueous phase of the bee pollen adhesive without the oily phase, the droplet samples on the silicon wafer were washed twice by 100 ml of fresh toluene for 1 min, followed by drying in a fume hood for 30 min to remove the residue toluene on the silicon wafer. Prepared samples were stored in chambers at different relative humidity levels of 15, 35, 57, and 75% RH for 24 h. Humidity in the chambers was controlled by solid salt (15% RH—calcium chloride) and supersaturated salt solutions (35% RH—calcium chloride, 57% RH—calcium chloride and sodium chloride, and 75% RH—sodium chloride). Three humidity-conditioned bee pollen adhesive droplets (with similar thicknesses in the range 1.5–1.8 μm) in each of the humidity chambers were chosen for the adhesion measurements. The thicknesses of the droplets were measured by an atomic force microscope (AFM; Veeco Dimension 3100, Santa Barbara, CA) with pyramidal-tipped scanning cantilevers (tapping mode, 0.75 Hz scanning rate) (ACTA, AppNano Inc., Santa Clara, CA).

**Solvent extraction.** The two liquid phases in the bee pollen adhesive were extracted and separated by a solvent extraction with toluene and water, referred to here as toluene extract and aqueous extract, respectively. A 2 g quantity of the dandelion pollen grains was dispersed in 20 ml of toluene, and the solution was shaken gently for 30 s. Most of the pollen grains settled on the bottom of the container within 30 s after shaking stopped. Only the toluene solution (orange color) was carefully transferred to another container by syringes with syringe filters (1 μm, 200 nm). Twenty milliliters of water were added to the toluene-washed pollen grains remaining in the container, and the pollen grains were dispersed in the water for 60 s. The pollen grains dispersed in the water solution were filtered by syringes with syringe filters (1 μm, 200 nm) to collect the water solution. Each solution was moved to a vacuum chamber, and the solvents (water and toluene) were evaporated under vacuum.

**Confocal fluorescence microscopy.** Bee pollen adhesive droplets were generated on a glass microscope slide via the same method as the sample preparation on the silicon wafer. Confocal images of the droplet samples were obtained using a Zeiss LSM 700 (Thornwood, NY) confocal microscope with FITC channels at 488 nm excitation, 20 °C and 40 ± 5% RH.

**Confocal Raman spectroscopy**. Raman spectra of samples were obtained using a Thermo Scientific Nicolet Almega XR spectrometer (Waltham, MA) with a 488 nm laser excitation source at 20 °C (40 ± 5% RH), and 10 scans were obtained for each sample. The laser beam was focused on the samples by using a ×50 objective. A confocal aperture of 100 μm pinhole was chosen, and the estimated spot size was 0.7 μm.

**Physical property characterization**. The density of both extracted phases was estimated by measuring the mass of each phase in a known volume, and the surface tension of both phases was measured using a ramé-hart automated goniometer (290-G1, Succasunna, NJ) with DROPimage Advanced software at 20 °C and at 57% RH. For determining the viscosity of the aqueous extract, three aqueous extract samples (1 ml) were stored in different relative humidity chambers (30, 57, 75% RH) for three days. The viscosity of these samples was measured as a function of shear rate in an MCR 302 rheometer (Anton Paar, Austria), using a cone-plate geometry (diameter 25 mm, cone angle 2°, sample volume 0.140 ml). The sample was kept at 20 °C via Peltier temperature control units on the rheometer; at the time of the measurements, the RH of the room was 40 ± 5%, but measurements were fairly quick (~30 min) and the sample was exposed to the atmosphere only at the rim of cone-plate geometry. Therefore, water uptake or loss during the course of testing could be neglected, as was shown by repeated viscosity measurements on the same sample.

**Adhesion measurement**. Adhesion of a sunflower pollen (*Helianthus annuus*) (Greer Laboratories, Lenoir, NC) on bee pollen adhesive was measured by colloidal probe microscopy by using the same AFM described above. Sunflower pollen was utilized in adhesion force measurement of bee pollen adhesive obtained from dandelion pollen grains. They have much more uniformly distributed surface features than dandelion[5], and that allows precise control over the wetting volume of fluid for wet adhesion measurement. The dandelion pollen was chosen simply because it is known to carry a large quantity of pollenkitt[5]. The pollenkitt of sunflower and dandelion pollen are known to have very similar surface tensions and composition due to their derivation from the same family (*Asteraceae*)[5]. To fabricate sunflower pollen probes, native non-defatted sunflower pollen grains were washed by an organic solvent mixture, as described in detail elsewhere[44]. One of the washed sunflower pollen grains was attached to a rectangular tipless AFM cantilever (ACST-TL, AppNano Inc., Santa Clara, CA), which has a nominal spring constant of 7.8 N m$^{-1}$, by a small amount of epoxy glue using a procedure described in detail elsewhere[45]. The spring constants of the fabricated cantilevers were determined according to the thermal tune method described by Hutter[46], incorporated in the Veeco AFM program. The approaching velocity was maintained at a small value of 500 nm s$^{-1}$, in order to minimize the viscous resistance to the pollen probes penetrating into the liquid droplet samples. The retraction rates of the probes were manipulated in the range of 0.5–117 μm s$^{-1}$ in order to probe both static and hydrodynamic contributions to capillary adhesion. All force measurements were performed under normal air condition (RH of 35–45%), and the loading forces on the cantilevers were controlled at 500 nN in all experiments.

## Data availability
All data are available from the corresponding author upon reasonable request.

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

## Acknowledgements

We acknowledge funding from Air Force Office of Scientific Research (Grant # FA9550-10-1-0555). In addition, David Hu and Marguerite Matherne of Georgia Tech contributed useful discussions about the relevance to natural adhesion in bees.

## Author contributions

D.S. and J.C.M. conceived the idea and designed the experiments. D.S. and J.C.M. wrote the article with the help of W.T.C. and V.B. D.S., W.T.C., Z.Q., and V.B. performed the experiments. D.S., J.C.M., W.T.C, and H.L. contributed to analysis and discussion of the results. J.C.M. supervised the project.

## Additional information

**Competing interests:** The authors declare no competing interests.

