## [Peer Review File · Nature Communications]

Reviewer #1 (Remarks to the Author):

This paper reports on some interesting properties of the adhesive holding together pollen pellets collected by honeybees. The authors find that the adhesive is two-phasic, consisting of oily plant-derived pollenkitt and an aqueous sugar-based compound produced by the bees. It is found that the viscosity of the sugar-based compound is humidity-dependent due to its hygroscopic nature, and its adhesion is therefore both rate- and humidity-dependent. However, the oily component strongly reduces water loss and uptake, thereby reducing the mixture's humidity-dependence of adhesion.

The reported effects are interesting and the experiments are clearly presented. However, I have some doubts about their biological relevance:

1. The „bee pollen“ used in this study was obtained from a commercial supplier – this is problematic for several reasons: First, it is unlikely that the pollen is only from a single plant species, i.e. other types of pollen will be present in the mixture. Second, the bees may add further compounds to the pollen in the hive, altering the properties from those in the pollen basket. Third, it is probably unclear how long the bee pollen was stored after collection. Commercial bee pollen is usually collected from bee hives, i.e. even if the collection time is known, the pollen may already have been present in the hive for a while. Long storage times may alter the physical and chemical properties of the pollen adhesive in several ways. Parts of the adhesive may evaporate. Moreover, pollen pellets are highly nutritious, and it is likely that some microbial digestion will take place (unless the pollen is completely dried or frozen).

A better way to obtain controlled pollen adhesive samples would be to collect fresh pollen baskets directly from honeybee workers foraging from a single plant species.

2. It is assumed that the liquid leaking from bee pollen samples is identical to the substance that holds together the pollen grains, but this has not been shown; both mixtures may well be different.

3. The study measures forces between pollenkitt-free sunflower pollen grains and a silicon wafer in the presence of the liquid collected from dandelion bee pollen, but more naturally relevant would be the adhesion between two dandelion pollen grains (with pollenkitt) in the presence of bee adhesive.

4. It is unclear how relevant rate-dependent forces are for holding together pollen grains in a foraging honeybee. Clearly, not only rate-dependent but also static adhesion forces are required. The reference to the high flight speeds of bees is slightly misleading in this context, as these would not directly translate to fast separation between pollen grains.

Specific comments

p.3 (and throughout paper) „pollens“

„pollen“ does not have a plural – perhaps replace by „pollen grains“

p.4 „pb-adhesive“

The fluid emanating spontaneously from bee pollen pellet is called here „pollen basket adhesive“, but I am not convinced that this liquid is identical to the substance that holds together the pollen pellets (there could well be components which do not transfer to the glass substrate).

p. 5 „We measured the wet adhesion of sunflower pollen (used as a control due to its regular surface features)“

In what sense is this a control? Why wasn't dandelion pollen used?

p.6, Fig.1

What do fresh dandelion pollen grains look like? Do they look different from the washed grain in Fig.1c?

p.6, Fig.1e

How many droplets were studied using Raman spectroscopy and fluorescence microscopy? Was the result always the same, i.e. did all the droplets have a hydrophilic core and a hydrophobic surrounding region? Did all watery droplets contain colloidal droplets with beta-carotene fluorescence?

p.7, „We hypothesize that the surrounding yellow region is pollenkitt, a plant-based lipid oil coat found on almost all entomophilous pollen.“

Why didn't the authors collect a fresh dandelion pollen sample? This would have provided a clear pollenkitt-only control, i.e. without bee adhesive. A comparison of 1) fresh dandelion pollen and 2) dandelion pollen from a (fresh) pollen basket would provide much clearer evidence of the two-phasic composition and the conclusion that the oily phase is plant-derived and the aqueous one is produced by the bees. Without such a sample, the conclusions of this paper are unnecessarily indirect.

p.9, Fig.2

What was the thickness of the core and surrounding regions for the measurements shown in this figure? I.e. how did the washing affect the thickness of the droplet? In Fig.2b, was there actually any liquid in the surrounding region, i.e. was the force curve different from the background (i.e. on the silicone wafer without a droplet)?

p.9, Fig.2a

The surrounding region, according to Fig.1f, is only ca 5 μ m wide – how could it be ensured that the sunflower pollen grains (with 30 μ m diameter) only made contact to this region?

p.9, Fig.2b

At what relative humidity was this plot recorded? Measurements from how many droplets are shown here?

p.9, Fig.2d

Are these values from the same droplet(s?) as those shown in Fig.2b?

p.10

The force measurements test a situation which is only partly representative of the natural function of pollen adhesive. Here, adhesion is measured between pollenkitt-free sunflower pollen grains and a silicon wafer, in the presence of a mixture of dandelion pollenkitt and bee adhesive. It would be better to measure the adhesion between two dandelion pollen grains (with pollenkitt) in the presence of bee adhesive.

p.10 „dandelion pollen has somewhat irregular features, making it difficult to reproducibly probe capillary adhesion of a single particle.“

What is meant by „irregular features“ (isn't sunflower pollen also irregular?) and why has this made a capillary adhesion measurement impossible?

p.11

This section assumes that the fluid in the core region is the same as the aqueous extract, and the fluid in the surrounding region the same as the toluene extract. This is not necessarily true – the composition and properties of the two fluid phases and the two extracts could be different, particularly if the adhesive contains solid parts that are absent from the collected liquid samples.

p.11

It would be helpful to check whether the „surrounding region“ changes its adhesive properties after washing off the oily phase

p.22 „To isolate the aqueous phase of the pb-adhesive without the oily phase, droplet samples were immersed in toluene for 1 min, followed by drying in a fume hood for 30 min“

It is not clear what happens to the oily compounds dissolved in toluene – do they evaporate with the toluene, or are less volatile compounds left behind?

„drying in a fume hood“ is slightly confusing here, as only toluene but not water was evaporated.

p.22 „droplets were measured by an atomic force microscope“

Using which mode (e.g. contact, tapping)? Was the shape of the droplets modified by the AFM investigation?

p.22 Solvent extraction

How many such samples were collected and studied in this way? Were the two fluid phases obtained immiscible?

p.24 „sunflower pollens were washed by an organic solvent mixture“

Did this solvent mixture remove the lipids?

p.24 „The spring constants of the fabricated cantilevers were determined according to the method described by Burnham and Hutter“

Which method? The cited paper Burnham et al. 2003 is a comparison of different calibration methods.

Reviewer #2 (Remarks to the Author):

This submission is well-conceived, complete, and timely. It is highly effective in that it pursues within a common experimental structure a dual goal of (i) describing the performance of a natural adhesive system of some importance alongside (ii) identifying features of the system that have potential technological significance.

The individual aspects of the study are all convincing. In an organized manner, the submission establishes

1) that the pollen-basket adhesive is made of distinct, immiscible chemical ingredients,

2) that these distinct sugary and oily phases possess different adhesion capacities (sugary > oily),

3) at the adhesion produced by the sugary phase exhibits a marked dependence on detachment rate,

4) that the adhesion produced by the sugary phase exhibits a marked dependence on environmental humidity (including a hump at intermediate RH levels), and

5) that the presence of the oily phase moderates the humidity dependence.

The extraneous variables encountered during these experiments are apparently well controlled; great care is taken when selecting pollen specimens and adhesive patches. Ancillary/follow-up experiments on water uptake, contact angle, and viscosity round out the main results. The authors also make an effort to put the results into physical context by establishing the equations that govern the various performance aspects of the adhesive and applying them to their data; these efforts are overall successful.

Some issues:

1) I feel that the viscosity measurements and analysis are of almost equal significance to the main adhesion measurements. Can these be included in the abstract?

2) Some of the experimental results are reported with uncertainties of two significant figures. Is there a reason for this or can they be made uniform at 1 sig. fig. throughout?

3) There is very little discussion here when it comes to identifying potential applications for this work. I recommend investing some thought to expand on this idea

Responses to Reviewers' comments

In what follows the referees' comments are in black and the authors' responses are in red.

Reviewer #1 (Remarks to the Author):

This paper reports on some interesting properties of the adhesive holding together pollen pellets collected by honeybees. The authors find that the adhesive is two-phasic, consisting of oily plant-derived pollenkitt and an aqueous sugar-based compound produced by the bees. It is found that the viscosity of the sugar-based compound is humidity-dependent due to its hygroscopic nature, and its adhesion is therefore both rate- and humidity-dependent. However, the oily component strongly reduces water loss and uptake, thereby reducing the mixture's humidity-dependence of adhesion.

The reported effects are interesting and the experiments are clearly presented. However, I have some doubts about their biological relevance:

1. The "bee pollen" used in this study was obtained from a commercial supplier – this is problematic for several reasons: First, it is unlikely that the pollen is only from a single plant species, i.e. other types of pollen will be present in the mixture. Second, the bees may add further compounds to the pollen in the hive, altering the properties from those in the pollen basket. Third, it is probably unclear how long the bee pollen was stored after collection. Commercial bee pollen is usually collected from bee hives, i.e. even if the collection time is known, the pollen may already have been present in the hive for a while. Long storage times may alter the physical and chemical properties of the pollen adhesive in several ways. Parts of the adhesive may evaporate. Moreover, pollen pellets are highly nutritious, and it is likely that some microbial digestion will take place (unless the pollen is completely dried or frozen). A better way to obtain controlled pollen adhesive samples would be to collect fresh pollen baskets directly from honeybee workers foraging from a single plant species.

These are all relevant questions. It is true that our study does not use freshly collected samples. However, we were very deliberate in the selection of a supplier (Greer Laboratories) and the type of sample purchased so that the collection, analysis of contaminants, and storage conditions are very well defined. For example, Greer Laboratories is a biopharmaceutical company specializing in the development of certified allergy immunotherapy products. According to the certificate of analysis (COA) provided by the supplier, these pollen grains are collected from bees in a manner that guarantees 97.2% purity from foreign pollen, plant parts, mold spores and rust. The COA is appended below, which shows a collection date and analysis data on purity. The grain purity was analyzed by the supplier via electron microscope at 400x and/or 1000x magnification, examining a minimum of 1000 particles. The amount of plant parts

present is 2.8% and the contamination by foreign pollen, mold or rust is 0.01%. For storage, the collected pollen grains were dried to 0.5% moisture content, and stored at < 0 °C. When we receive the pollen, they are stored under dry conditions for a short amount of time (weeks at most) at 20 °C until use. Most of the experiments performed in this study were carried out in early 2016, just after receipt of the samples from Greer. The pollens are certified for a use lifetime of 10 years for their usual use in pharmaceutical development, i.e., this is the timescale over which Greer guarantees chemical stability of the pharmaceutically-useful compounds. ***We describe the details of our pollen samples in page 22.***

Because we use a sample from nature that is stored, albeit carefully and under well-defined conditions, we must stop short of drawing conclusions about the adhesive activity in the natural world. ***We have gone back through the paper to be sure to clarify this point (page 8 and 21). Rather, our paper has focused on the novel adhesive properties of a two-phase system taken from nature, and the most important conclusions are in the materials and adhesion novelty.*** This is achieved by using biological samples collected and stored under well-characterized conditions. As the reviewer suggests, we agree that using freshly-captured samples would be an obvious and crucial next step in confirming that the adhesive phenomena we uncovered here are also actively occurring in natural systems, e.g., during pollination or pollen collection. There are certain biological parameters that could be controlled – indeed it is possible that entering a hive could cause chemical changes. We have pointed out some of these potential caveats in the paper, highlighted as changes. Our study provides ample evidence that this question should be investigated, and our aim is that it will inspire those studies. However, even when using freshly-collected samples, one is going to encounter the same questions about evaporation of water and organics and chemical reactions, but on shorter timescales. The equilibration of water is fast enough (hours for water vapor equilibrium) that even a naturally-collected sample brought immediately to the lab would be changing during the time required to do adhesion measurements (hours and usually done over a few days), adapting to the laboratory's humidity level (the problems would not be eliminated, but they would be different).

Certificate of Analysis

STALLERGENES  GREER

P.O. Box 800 Lenoir, NC 28645, USA
T: 800.378.3907 or 828.759.7859 F: 828.754.0616

COMMON NAME: DANDELION
BOTANICAL NAME: *TARAXACUM OFFICINALE*
FAMILY NAME: ASTERACEAE

Item Number
Collection Date
Lot Number

COLLECTION AREA

Country
State

ORIGIN OF PLANT

- Wild
 Cultivated
 Unknown

COLLECTION METHOD

- Vacuum
 Water Set
 Cut, Dried and Sieved
 Other

CLEANING METHOD

- Mechanically sieved
 Not chemically washed
 Chemically washed

DRYING

% Moisture Pollen meets moisture specifications of <=4.5%
Rust/Smuts meets moisture specifications of <=9.0%

STORAGE CONDITIONS

- Stored at < 0 °C

Best If Used By

IDENTITY AND PURITY

Grain Size μ

Results of microscopic examination at 400 and/or 1000 magnification, examining a minimum of 1000 pollen grains or smut/rust spores

Plant Parts %

Anemophilous plants - <5%
Entomophilous plants - <12%

Contamination %

(Foreign pollen, mold spores, smut spores, rust spores - <1%)

Macroscopical ID

COMMENTS

We hereby certify that the pollen/rust/smud has been processed as indicated and has been microscopically examined for identity and purity.

Signature/Date: _____

2. It is assumed that the liquid leaking from bee pollen samples is identical to the substance that holds together the pollen grains, but this has not been shown; both mixtures may well be different.

The reviewer asks an important question here: whether the fluid between pollens in a basket (as it is carried on a bee in nature) is identical to what we collect from the pollens. We used two techniques to capture fluid from the pollen: by gravity-induced drainage and by solvent extraction. Based on surface science and phase behavior there is little plausible reason to believe that the fluid that drains onto silicon is different. The wetting of the fluids on silicon may well be different than in contact with sporopollenin. However, the compositions and phases present should not change because of these wetting differences (there's no plausible effect that would induce changes in the phase compositions simply by exposure to a different substrate, unless the substrate surface area were orders of magnitude larger, as in a chromatography column for example). It is possible that more of the oil phase versus the water phase may drain, leading to some quantitative differences in the volumes of the two fluids. However, this would not affect the phase internal compositions, surface tensions, or the oil spreading over the water phase. In summary, the silicon can affect the contact angle and spreading rate of the droplets, but not (at these drop sizes) the composition or the presence of multiple phases, nor their relative spreading on one another (oil phase spreading over water phase). ***We have added comment and discussion about these important points (Page 8 and 21).***

3. The study measures forces between pollenkitt-free sunflower pollen grains and a silicon wafer in the presence of the liquid collected from dandelion bee pollen, but more naturally relevant would be the adhesion between two dandelion pollen grains (with pollenkitt) in the presence of bee adhesive.

The reviewer suggests using unwashed dandelion pollen for the substrate (instead of pb-adhesive droplet on the flat surface) and colloidal probe (instead of sunflower pollen). We chose to use sunflower grains because of the regular features they possess, and that the approach and retraction are easier to replicate over many samples when we have a relatively large single spine (as is present on sunflower) interacting with a flat substrate. A pollen-pollen experiment like that proposed by the reviewer is a good idea for a follow-up study, but for understanding the properties of the fluid extract alone, we chose the use of a flat substrate as is the standard in colloidal probe studies. We've carried out pollen-stigma studies (published Lin, H., Qu, Z. & Meredith, J. C. Pressure Sensitive Microparticle Adhesion through Biomimicry of the Pollen-Stigma Interaction. *Soft Matter* 12, 2965–2975 (2016)) as well as pollen-pollen studies

(unpublished) that indicate the confounding solid-solid (friction and adhesion) interactions that occur when both surfaces are structured (even when dry).

For reproducible measurement of colloidal probe microscope (CPM) for a wet substrate, the substrate structure and the thickness of the pb-adhesive must be controlled. The dandelion pollen (radius around 15 μm) has irregular concave structures (diameter between 5-10 μm) with short spines on the surface as shown in Extra Supplementary Fig. 1. These surface features will create a large uncertainty for the wet adhesion measurement because it will be difficult to control the orientation of the interactions as well as the number of spines that touch (because the spines are relatively short). On the other hand, the sunflower pollen has spines that are nearly identical in size (spine length, 3-4 μm , spine tip radius $125 \pm 10 \text{ nm}$) that are distributed uniformly (number of spines per unit area, $0.32 \mu\text{m}^{-2}$). Thus, the sunflower represents a compromise of using sporopollenin from a pollen within the same family (*Asteraceae*) as dandelion but with much larger and more uniform features. With the controlled film thickness, the sunflower pollen grains allow precise control over the wetting volume of fluid for wet adhesion measurement.

Therefore, we used flat and hard silicon wafer with pb-adhesive coating instead dandelion pollen as a substrate and sunflower colloidal probe instead of dandelion probe. We had reproducible adhesion magnitudes for different pb-adhesive droplet substrates which have similar adhesive film thickness. The introduction of the dandelion pollen interactions would introduce a complex orientation-dependent geometrical interaction that would make our force measurements difficult if not impossible to interpret. So, this flat-surface study is critical to first understand the fluid properties prior to going to the next level of pollen-pollen interactions. The measurement of the exact adhesion between the dandelion pollens was not our main interest, but the pb-adhesive property and its changes under the relative humidity.

Extra Supplementary Figure 1 | SEM images of a dandelion pollen. (a) Before the solvent wash, the surface of the pollen is covered by pb-adhesive. (b) After the solvent wash (water and toluene), the surface liquid coating was removed. The red arrows in the image indicate the irregular concave structures on the washed dandelion pollen.

4. It is unclear how relevant rate-dependent forces are for holding together pollen grains in a foraging honeybee. Clearly, not only rate-dependent but also static adhesion forces are required. The reference to the high flight speeds of bees is slightly misleading in this context, as these would not directly translate to fast separation between pollen grains.

Capillary forces (static adhesion) and viscous dissipation (rate-dependent adhesion) are the two main contributors to the wet adhesive force, and the viscous force becomes the main contributor when the liquid capillary is a viscous liquid. We didn't eliminate or discount the contribution of the static adhesion force, but both static and dynamic adhesion were measured and estimated via experimental study and a theoretical model as shown in Fig. 4 and Table 1. We claimed that the viscous dissipation (hydrodynamic capillary force) was overwhelming the static capillary force under dynamic condition above a critical retraction rate. We observed the pb-adhesive is a highly viscous liquid. (It's four orders of magnitude higher viscosity than water), and the estimated static capillary forces (Table 1) were two orders of magnitude lower than the viscous force. ***We agree that the flight speed could be misinterpreted and have decided to remove that from the paper (page 3).***

Specific comments

p.3 (and throughout paper) „pollens“

„pollen“ does not have a plural – perhaps replace by „pollen grains“

We changed the manuscript with this rule: when referring to specific samples of pollen grains utilized only in this work, we utilize “pollen grains”, but otherwise refer to general plural pollen as “pollen” when referring to their occurrence in nature. ***We have checked the manuscript and have changed any instances accordingly.***

p.4 „pb-adhesive“

The fluid emanating spontaneously from bee pollen pellet is called here “pollen basket adhesive”, but I am not convinced that this liquid is identical to the substance that holds together the pollen pellets (there could well be components which do not transfer to the glass substrate).

We observe the humidity-dependent physical property of the liquids we removed from pollens from both liquid that drains under gravity onto glass and the liquid that is extracted by solvent extraction. The former method is preferred as it does not introduce the selective influence of a solvent, but the latter is more likely to remove all substances from the pollens. We felt that using two different removal methods and comparing them was a careful approach. Still, one can argue that our use of the word pb-adhesive should distinguish explicitly from the liquids present in the natural state prior to their removal. As shown in Extra Supplementary Fig. 1, most of the adhesive component on the dandelion pollen are extracted based on the SEM images, and the humidity-tolerant property and rate-dependent adhesive properties are shown from both glass transferred and solvent extracted samples, and both samples have the identical Raman spectrum as shown in Extra Supplementary Fig. 2. We understand the reviewer's concern that the liquid in pollen basket might not be absolutely identical to the transferred and extracted liquid. However, we clearly showed the presence and function of the two phases from the glass transferred (core and surrounding regions in figure 1 and 2) and solvent extracted (aqueous and oily phases in figure 3 and 4) samples. This evidence indicates that the two phases are the main components of the liquid adhesive of the pollen pellet and, the liquid adhesive has the humidity-tolerant and rate-dependent adhesive properties.

Extra Supplementary Figure 2 | Confocal Raman spectra of Pb adhesive droplet and extracted phases. (a) The red line and the blue line represent the surrounding and the core regions of the droplet deposited on a Si wafer. (b) The orange line and black line represent the oil and aqueous phases extracted by water and toluene solvent.

p. 5 „We measured the wet adhesion of sunflower pollen (used as a control due to its regular

surface features)“

In what sense is this a control? Why wasn't dandelion pollen used?

The word 'control' was a poor choice in the manuscript. Indeed, it is not a control experiment. However, the use of sunflower was chosen to allow better control of the orientation of the probe and liquid than would be possible with the dandelion pollen. The dandelion pollen (radius around 15 μm) has irregular concave structures (diameter between 5-10 μm) with short spines on the surface. The redundant surface features create a large error for the wet adhesion measurement. We add the larger image of the pollen in Extra Supplementary Fig. 1. Otherwise, the sunflower pollen has uniformly distributed surface feature. With the controlled film thickness, the sunflower pollen grains allow precise control over the wetting volume of fluid for wet adhesion measurement. Extra Supplementary Fig. 3 shows that the adhesion of the natural dandelion pollen *collected directly from plants* (coated with pollenkitt only, not bee-collected pollen), and shows the highest error bar among the different pollen species because of the surface feature and orientation of dandelion pollen.

Extra Supplementary Figure 3 | Adhesion force between natural pollen and various substrates. The purple bar represents the adhesion of the dandelion pollen, and it shows the highest error bar for dandelion surface feature and orientation. The error bars are 95% confidence intervals. (This figure is captured from Lin, H., Gomez, I. & Meredith, J. C.

Pollenkitt wetting mechanism enables species-specific tunable pollen adhesion. *Langmuir* 29, 3012–3023 (2013).

p.6, Fig.1

What do fresh dandelion pollen grains look like? Do they look different from the washed grain in Fig.1c?

They look very different before and after washing and the removal of the coating before and after is clear. It is a good idea to clarify this feature for readers. We add an Extra Supplementary Fig. 1 to present the difference clearly.

p.6, Fig.1e

How many droplets were studied using Raman spectroscopy and fluorescence microscopy? Was the result always the same, i.e. did all the droplets have a hydrophilic core and a hydrophobic surrounding region? Did all watery droplets contain colloidal droplets with beta-carotene fluorescence? ‘

We tested more than ten droplets with Raman spectroscopy and fluorescence microscopy and the results were consistent. After droplet sample preparation process described in Methods, we observed the prepared 165 droplets using optical microscope, and 140 droplets have a hydrophilic core and a hydrophobic surrounding region. Some droplets (about 15%) may not having the hydrophilic core, and it might attribute to the uneven distribution during the fluid spreading on glass surface or air blowing process to remove the pollen grains. We picked the droplets with over 20 μm width for the confocal Raman spectroscopy, and the peaks for the carotenoids were observed for all droplets.

p.7, „We hypothesize that the surrounding yellow region is pollenkitt, a plant-based lipid oil coat found on almost all entomophilous pollen.“

Why didn't the authors collect a fresh dandelion pollen sample? This would have provided a clear pollenkitt-only control, i.e. without bee adhesive. A comparison of 1) fresh dandelion pollen and 2) dandelion pollen from a (fresh) pollen basket would provide much clearer evidence of the two-phasic composition and the conclusion that the oily phase is plant-derived and the aqueous one is produced by the bees. Without such a sample, the conclusions of this paper are unnecessarily indirect.

The native dandelion pollen samples (pollenkitt-only control) were investigated in our previous study (Lin, H., Gomez, I. & Meredith, J. C. Pollenkitt wetting mechanism enables species-specific tunable pollen adhesion. *Langmuir* 29, 3012–3023 (2013)). As shown in Extra Supplementary Fig. 4 (b), we didn't observe two-phasic composition, especially the core aqueous phase, from the

liquid phase residue of the native dandelion pollen sample after pollen removal. Regarding the source of the oil phase, the pollenkitt is a highly plausible one based on what is known about the composition and properties of pollenkitt, for example its loading with carotene-like compounds (as confirmed in Raman spectra) and similarity in adhesive profile with the prior study of pollenkitt adhesion (where samples were taken direct from plants). But, the reviewer is correct that we cannot confirm that the oil phase comes from the plant, only that this is very likely. **We have pointed this out in the revision (Page 2,8, and 21)**. Even without knowing the oil phase source, we know that it is there in the biological sample, and our primary point is that a naturally-derived material has these novel adhesive properties in combination with humidity.

Extra Supplementary Figure 4 | Pollenkitt deposition from native dandelion pollen on Si surface (a) SEM image of the native dandelion pollen on Si surface. The pollenkitt capillary bridges between dandelion pollen and Si are observed. (b) Optical microscope image showing residual dandelion pollenkitt, after pollen removal. (This figure is obtained from the reference 5 (Lin, H., Gomez, I. & Meredith, J. C. Pollenkitt wetting mechanism enables species-specific tunable pollen adhesion. *Langmuir* 29, 3012–3023 (2013))

p.9, Fig.2

What was the thickness of the core and surrounding regions for the measurements shown in this figure? I.e. how did the washing affect the thickness of the droplet? In Fig.2b, was there actually any liquid in the surrounding region, i.e. was the force curve different from the background (i.e. on the silicone wafer without a droplet)?

The thickness of the core and surrounding regions were estimated using the jump-in distance of the approach curves shown in supplementary Fig.1. For the precise control over the wetting volume of fluid for wet adhesion measurement, the core and surrounding regions with similar thickness (0.8-1.1 μm) were chosen for the measurements in Fig. 2b. The washing could affect the thickness of the droplets because the measured interfacial tension of the oily phase in the

air was 22.5 mN/m and the aqueous phase in the air was 52.5 mN/m as shown in Supplementary Table 1. The thicknesses of the adhesive droplets in Fig. 2c and d were directly measured before the all force measurements as shown in Supplementary Fig.2 and 4.

The approach and the retraction curves in Supplementary Fig.1 show a typical wet adhesion force-distance curve shape. For example, the jump-in distance of the dry adhesion is less than 10 nm if the governing force of the pollen adhesion on the silicon wafer is a dry adhesion (mainly contributed by the vdW interaction), but it was about 800 nm according to the measurement. We are very confident that the adhesive force on the surrounding region is governed by wet adhesion.

p.9, Fig.2a

The surrounding region, according to Fig.1f, is only ca 5 μ m wide – how could it be ensured that the sunflower pollen grains (with 30 μ m diameter) only made contact to this region?

The droplet in Fig. 1f is a sample on microscope glass slide for the confocal fluorescence microscopy. For the force measurement of surrounding region, we picked a larger droplet with a larger surrounding area to make sure that the core phase is not involved in the adhesive force measurement. In addition, the AFM instrument has an in-built high-resolution microscope, so it was not challenging to control the contact location of the sunflower pollen.

p.9, Fig.2b

At what relative humidity was this plot recorded? Measurements from how many droplets are shown here?

The experiment was performed at 40% RH, and the measurements were from 3 different droplets which have a similar film thickness in the range a 0.8-1.1 μ m. ***The first paragraph in page 10 describes the experimental condition for this measurement.***

p.9, Fig.2d

Are these values from the same droplet(s?) as those shown in Fig.2b?

These are different droplets from Fig. 2b. The droplets for Fig. 2c and Fig. 2d were kept at the controlled relative humidity (15%, 35%, 57%, 75% RH) for 24 h, and the details of the droplet preparation is described in the first part of Methods section.

p.10

The force measurements test a situation which is only partly representative of the natural function of pollen adhesive. Here, adhesion is measured between pollenkitt-free sunflower pollen grains and a silicon wafer, in the presence of a mixture of dandelion pollenkitt and bee

adhesive. It would be better to measure the adhesion between two dandelion pollen grains (with pollenkitt) in the presence of bee adhesive.

This question is addressed with the previous question 3 (page 4 and 5 in this response) in the beginning of the review where the same comment was made.

p.10 „dandelion pollen has somewhat irregular features, making it difficult to reproducibly probe capillary adhesion of a single particle.“

What is meant by „irregular features“ (isn't sunflower pollen also irregular?) and why has this made a capillary adhesion measurement impossible?

This question is addressed with the previous question 3 (page 4 and 5 in this response) in the beginning of the review where the same comment was made.

p.11

This section assumes that the fluid in the core region is the same as the aqueous extract, and the fluid in the surrounding region the same as the toluene extract. This is not necessarily true – the composition and properties of the two fluid phases and the two extracts could be different, particularly if the adhesive contains solid parts that are absent from the collected liquid samples.

In this context, our assumption was based on the Confocal Raman spectroscopy spectra of both core and surrounding regions and the extracted aqueous and oily phases, shown in Fig. 1e.

During the extraction process, it might be possible minor components of aqueous phase could be missed in extracted aqueous phase or minor components of oily phase might be missed in extracted oily phase. However, it is clear that the major aqueous phase of pb-adhesive would dissolve in polar solvent water and the major oily phase of pb-adhesive would be dissolved in non-polar solvent toluene, and the Confocal Raman spectroscopy spectra of extracted phases were almost same as the core and surrounding region as shown in Extra Supplementary Fig. 2. We do not exclude the possibility that one of the liquid phases may contain microscopic solid particulates, and that these may not transfer completely. However, we would expect that the solids that are already suspended in the pollen basket adhesive fluid would remain suspended during drainage off of the pollen. Most importantly, the main key (adhesive and humidity-tolerant properties) functions of the liquid phases are preserved for both the drained and solvent-extracted phases, Fig.3 and 4. We agree with the concern from the reviewer that we cannot exhaustively exclude that there are differences between the extracted aqueous and oily phases might not be identical to the aqueous and oily phase in the pb-adhesive droplet. **To clarify this, we rewrite the term 'ex-aqueous and ex-oily phases' for the aqueous and oily phases from solvent extraction in our contexts (page 14).**

p.11

It would be helpful to check whether the “surrounding region” changes its adhesive properties after washing off the oily phase

After the toluene washing, we observed that the surrounding regions were clearly washed away based on our optical microscope images shown in Fig. 2c and d. The AFM cross-sectional images of the droplets before (Supplementary Fig.4) and after washing (Supplementary Fig.2) also support the observation. While it would be helpful to check this with adhesion measurements outside the core after drying, this is no longer a possibility for the samples in question.

p.22 „To isolate the aqueous phase of the pb-adhesive without the oily phase, droplet samples were immersed in toluene for 1 min, followed by drying in a fume hood for 30 min“

It is not clear what happens to the oily compounds dissolved in toluene – do they evaporate with the toluene, or are less volatile compounds left behind?

„drying in a fume hood“ is slightly confusing here, as only toluene but not water was evaporated.

The purpose of this washing process is to remove the oily phase coating on the aqueous phase of the pb-adhesive droplet. The pb-adhesive droplet samples, which are drained from 35mg of dandelion pollen, were generated on the silicon wafer. The droplet samples were washed in the two of 100 ml fresh toluene for 30 sec each. (After the first 30 sec washing, the surrounding regions are not observed via optical microscope, and the additional 30 sec washing process was added to make sure that the all surrounding regions are removed.) Since we used the sufficient amount of the washing solvent (100ml), which is much higher than surrounding region volume, there are aqueous phase droplets and the small amount of toluene left on the silicon wafer after the washing. The drying process in a fume hood was performed to remove the residue toluene on the silicon wafer. In the ambient condition, the small amount of the toluene was fully evaporated in a short time, but the water in the aqueous phase was not evaporated because of the water absorption property of the aqueous phase. Rather, water in the aqueous phase equilibrates with water vapor in the environment at the relatively humidity used for testing. **We clarify this washing process by adding the details on page 22.**

p..22 „droplets were measured by an atomic force microscope“

Using which mode (e.g. contact, tapping)? Was the shape of the droplets modified by the AFM investigation?

The droplet was measured by tapping mode as described in page 22. We haven't seen any detectable droplet shape changes during the measurement. The dimension of AFM cantilever tip (6 nm, radius) was much smaller than the droplets, and the droplets were a viscous liquid.

p.22 Solvent extraction

How many such samples were collected and studied in this way? Were the two fluid phases obtained immiscible?

For every extraction, we used about 2 g of dandelion pollen. For this study, we consumed more than 50 g of the pollen followed this extraction process. After extraction, the extracted aqueous and oily phases were immiscible, and the oily phase coated the air-aqueous phase interface.

p.24 „sunflower pollens were washed by an organic solvent mixture“

Did this solvent mixture remove the lipids?

When we receive the native pollen (non-defatted) grains, they were coated with the lipid oil (pollenkitt). We washed away this lipid oil coat, and it allows precise control over the wetting volume of fluid for wet adhesion measurement with the controlled film thickness.

p.24 „The spring constants of the fabricated cantilevers were determined according to the method described by Burnham and Hutter“

Which method? The cited paper Burnham et al. 2003 is a comparison of different calibration methods.

We used the thermal tune method in the Veeco AFM system, and this method was originally proposed by Hutter. As the reviewer pointed out, the cited paper, Burnham et al. 2003 (reference 47), is not an appropriate reference for this particular method. **Therefore, we remove the citation 47 and rewrite the calibration details on page 24.**

Reviewer #2 (Remarks to the Author):

This submission is well-conceived, complete, and timely. It is highly effective in that it pursues within a common experimental structure a dual goal of (i) describing the performance of a natural adhesive system of some importance alongside (ii) identifying features of the system that have potential technological significance.

The individual aspects of the study are all convincing. In an organized manner, the submission establishes

- 1) that the pollen-basket adhesive is made of distinct, immiscible chemical ingredients,
- 2) that these distinct sugary and oily phases possess different adhesion capacities (sugary > oily),
- 3) at the adhesion produced by the sugary phase exhibits a marked dependence on detachment

rate,

4) that the adhesion produced by the sugary phase exhibits a marked dependence on environmental humidity (including a hump at intermediate RH levels), and

5) that the presence of the oily phase moderates the humidity dependence.

The extraneous variables encountered during these experiments are apparently well controlled; great care is taken when selecting pollen specimens and adhesive patches. Ancillary/follow-up experiments on water uptake, contact angle, and viscosity round out the main results. The authors also make an effort to put the results into physical context by establishing the equations that govern the various performance aspects of the adhesive and applying the to their data; these efforts are overall successful.

Some issues:

1) I feel that the viscosity measurements and analysis are of almost equal significance to the main adhesion measurements. Can these be included in the abstract?

Yes. We add the viscosity measurements and analysis part in our abstract as suggested. (Page 2)

2) Some of the experimental results are reported with uncertainties of two significant figures. Is there a reason for this or can they be made uniform at 1 sig. fig. throughout?

We accept the reviewer's suggestion and unify the significant figures of the experimental data. (Page 12 and 18)

3) There is very little discussion here when it comes to identifying potential applications for this work. I recommend investing some thought to expand on this idea.

We discuss the potential applications as suggested (page 20).

Reviewer #1 (Remarks to the Author):

This paper has been thoroughly revised and most questions have been answered convincingly.

My only remaining concern is that the use of commercial pollen still weakens the biological relevance of the findings presented in the paper, and in my view the conclusions aren't written in a way that makes this sufficiently clear. For example, the text states (p.4) "we investigate the adhesive properties properties of the liquid secretion of dandelion pollen collected from the pollen baskets (named as pb-adhesive in this paper) of honey bees", and the title includes "Pollen Basket Fluid", but this is not correct – the pollen samples were collected from bee hives, and their properties may well differ from those found in the bees' pollen baskets. Even if the pollen is only from a single plant species, it may contain further compounds added by the bees in the hive, altering the properties from those in the pollen basket. Even if the pollen samples were stored under well-controlled conditions after collection, the pollen was present in the hive for a while before collection, potentially affecting its properties.

The paper needs to be revised to make this clear and to remove misleading references (e.g. Fig.1b suggests that this is a sample from a pollen basket, but it probably isn't).

Minor points:

- Although the reply suggested that reference to the bees' flight speed has been removed, the sentence still seems to be there (line 363)

line 232 "The aqueous and oily phase fluid in pb-adhesive were extracted and separated using solvent extraction with toluene and water as described in Methods (named as ex-aqueous and ex-oily phase, respectively)"

The two new terms „ex-aqueous“ and „ex-oily“ do not help to increase the clarity. Why not simply call the samples "aqueous extract" and "toluene extract"?

To prove that the aqueous extract is similar to the fluid in the core region, and the toluene extract to the fluid in the surrounding region, it would be helpful to show the Raman data (extra Supplementary Figure 2) at least in the SI.

Reviewer #2 (Remarks to the Author):

I think that the re-submission is improved over the original (high-quality) submission, and that you have made a sober, effective effort to address reviewer concerns.

-JBP

REVIEWERS' COMMENTS:

Reviewer #1 (Remarks to the Author):

This paper has been thoroughly revised and most questions have been answered convincingly.

My only remaining concern is that the use of commercial pollen still weakens the biological relevance of the findings presented in the paper, and in my view the conclusions aren't written in a way that makes this sufficiently clear. For example, the text states (p.4) "we investigate the adhesive properties of the liquid secretion of dandelion pollen collected from the pollen baskets (named as pb-adhesive in this paper) of honey bees", and the title includes "Pollen Basket Fluid", but this is not correct – the pollen samples were collected from bee hives, and their properties may well differ from those found in the bees' pollen baskets. Even if the pollen is only from a single plant species, it may contain further compounds added by the bees in the hive, altering the properties from those in the pollen basket. Even if the pollen samples were stored under well-controlled conditions after collection, the pollen was present in the hive for a while before collection, potentially affecting its properties. The paper needs to be revised to make this clear and to remove misleading references (e.g. Fig.1b suggests that this is a sample from a pollen basket, but it probably isn't).

Based on communication with the supplier, we confirmed that the bee-collected pollen grains were not collected from the inside of hive, but the pollen grains are collected in traps at the hive entrance. When the bee passes through holes in the trap, pollen is scraped off the insect and based on known behavior, this pollen should have been packed into a basket prior to hive entry. However, we still accept that we cannot exclude the possibility that non-basket pollen or compounds from the hive potentially on the bee surface could enter the samples. But we do know that pollens are not intentionally taken from within the hive.

Therefore, we make four changes in our original manuscript. *First, the "pollen basket fluid" in the original title is replaced by "bee-collected pollen fluid". Second, we substitute the "pollen basket fluid" term in our text (p.4, p.6, p.9 and p.21) to other more generalized terms, such as "bee-collected pollen adhesive". Third, the "pollen basket" term in Fig. 1b and the arrow between Fig. 1a and Fig. 1b were removed to avoid the misleading information. Fourth, the "pb-adhesive" term is replaced by "bee pollen adhesive" in the entire text. Further we discuss in the results that hive entrance pollen traps have been used to focus on collecting the pollen baskets (p.7), but that we cannot exclude possible other compounds being introduced from the hive (carried on bee when they exit hive) or that single pollens not in a basket were collected (p.8).*

Minor points:

- Although the reply suggested that reference to the bees' flight speed has been removed,

the sentence still seems to be there (line 363)

We did remove the flight speed in the introduction, but we missed this discussion at the end of the paper. *We have revised this to discuss instead use of the rate dependence in synthetic adhesives; and we have suggested that in order to investigate any natural function of rate dependence that control experiments with fresh pollen need to be carried out (p.21).*

line 232 “The aqueous and oily phase fluid in pb-adhesive were extracted and separated using solvent extraction with toluene and water as described in Methods (named as ex-aqueous and ex-oily phase, respectively)” The two new terms „ex-aqueous“ and „ex-oily“ do not help to increase the clarity. Why not simply call the samples “aqueous extract” and “toluene extract“?

We accept the reviewer’s suggestion, *and the “ex-queous” and “ex-oily” are replaced by “aqueous extract” and “toluene extract” in the entire text.*

To prove that the aqueous extract is similar to the fluid in the core region, and the toluene extract to the fluid in the surrounding region, it would be helpful to show the Raman data (extra Supplementary Figure 2) at least in the SI.

We agree with this suggestion, *so the extra supplementary figure 2 in our initial response is added in SI as Supplementary figure 5 (p. 14).*

Reviewer #2 (Remarks to the Author):

I think that the re-submission is improved over the original (high-quality) submission, and that you have made a sober, effective effort to address reviewer concerns.

Thank you for your thoughtful suggestions that you made on our original submission.